# Diagnosis and Management of Mixed Transcortical Aphasia Due to Multiple Predisposing Factors, including Postpartum and Severe Inherited Thrombophilia, Affecting Multiple Cerebral Venous and Dural Sinus Thrombosis: Case Report and Literature Review

**DOI:** 10.3390/diagnostics11081425

**Published:** 2021-08-06

**Authors:** Dragoș Cătălin Jianu, Silviana Nina Jianu, Traian Flavius Dan, Nicoleta Iacob, Georgiana Munteanu, Andrei Gheorghe Marius Motoc, Adelina Băloi, Daniela Hodorogea, Any Docu Axelerad, Horia Pleș, Ligia Petrica, Anca Elena Gogu

**Affiliations:** 1Department of Neurology, “Victor Babeș” University of Medicine and Pharmacy, 300041 Timișoara, Romania; dcjianu@yahoo.com (D.C.J.); agogu@yahoo.com (A.E.G.); 2Centre for Cognitive Research in Neuropsychiatric Pathology (NeuroPsy-Cog), “Victor Babeș” University of Medicine and Pharmacy, 300736 Timișoara, Romania; ples.horia@umft.ro (H.P.); ligia_petrica@yahoo.co.uk (L.P.); 3First Department of Neurology, “Pius Brînzeu” Emergency County Hospital, 300736 Timișoara, Romania; daniela_hodorogea@yahoo.com; 4Department of Ophthalmology, “Dr. Victor Popescu” Military Emergency Hospital, 300080 Timișoara, Romania; silvianajianu@yahoo.com; 5Neuromed Diagnostic Imaging Centre, Department of Multidetector Computed Tomography and Magnetic Resonance Imaging, 300218 Timișoara, Romania; nicoiacob@yahoo.co.uk; 6Department of Anatomy and Embryology, “Victor Babeş” University of Medicine and Pharmacy, 300041 Timișoara, Romania; 7Department of Anaesthesia and Intensive Care, “Pius Brînzeu” Emergency County Hospital, 300736 Timișoara, Romania; adelina7289@gmail.com; 8Department of Neurology, General Medicine Faculty, Ovidius University, 900470 Constanța, Romania; docuaxi@yahoo.com; 9Department of Neurosurgery, “Victor Babeș” University of Medicine and Pharmacy, 300041 Timișoara, Romania; 10Department of Internal Medicine II, Division of Nephrology, “Victor Babeș” University of Medicine and Pharmacy, 300041 Timișoara, Romania

**Keywords:** cerebral venous and dural sinus thrombosis (CVT), puerperium, inherited thrombophilia, mixed transcortical aphasia, low-molecular-weight heparin (LMWH)

## Abstract

Cerebral venous and dural sinus thrombosis (CVT) is an uncommon disease in the general population, although it is a significant stroke type throughout pregnancy and the puerperium. Studies describing this subtype of CVT are limited. Most pregnancy-associated CVT happen in late pregnancy, or more commonly in the first postpartum weeks, being associated with venous thrombosis outside the nervous system. *Case presentation*: The current study describes a case of multiple CVT in a 38-year-old woman with multiple risk factors (including severe inherited thrombophilia and being in the puerperium period), presenting mixed transcortical aphasia (a rare type of aphasia) associated with right moderate hemiparesis and intracranial hypertension. The clinical diagnosis of CVT was confirmed by laboratory data and neuroimaging data from head computed tomography, magnetic resonance imaging, and magnetic resonance venography. She was successfully treated with low-molecular-weight heparin (anticoagulation) and osmotic diuretics (mannitol) for increased intracranial pressure and cerebral edema. At discharge, after 15 days of evolution, she presented a partial recovery, with anomic plus aphasia and mild right hemiparesis. Clinical and imaging follow-up was performed at 6 months after discharge; our patient presented normal language and mild right central facial paresis, with chronic left thalamic, caudate nucleus, and internal capsule infarcts and a partial recanalization of the dural sinuses.

## 1. Introduction

The occurrence of cerebral venous and dural sinus thrombosis (CVT) has been shown to be greater in persons with thrombophilia, patients under 40 years old, pregnant or postpartum women, and those who use oral contraceptives [1,2,3,4,5].

The most common symptoms are headache, papilledema, convulsions, and any of the focal neurological deficits, such as unilateral weakness of the facial muscles and limbs [1]. Due to its large clinical spectrum, CVT is frequently undiagnosed, and if not quickly treated can sometimes be a fatal condition [6]. Timely diagnosis (including with different neuroimaging techniques) and appropriate treatment can substantially improve outcomes for patients with CVT [1,2,3,4,5].

## 2. Case Report

A 38-year-old woman with pre-existing hypertension on her 18th postpartum day was admitted to the emergency department (ED) after sudden onset of severe (8/10) headache for two days, followed by moderate weakness of the right limbs several hours before the ambulance was called. She was also unable to understand orders and to speak properly, although at the same time she could repeat words and sentences. She has no history of fever or trauma, although the details of her obstetrical history were of great concern. This was her third pregnancy; the two earlier pregnancies had ended in first trimester miscarriages, with the last one ended in a preterm cesarean delivery at 30 weeks gestation for early severe preeclampsia 18 days before she was admitted to our hospital.

### 2.1. Clinical Presentation

In the general clinical examination, the patient was shown to be obese (BMI = 30), with moderate hypertension and tachycardia (BP = 150/100 mmHg, HR = 94 bpm) but without fever.

In the ophthalmological exam, she had visual acuity of 20/20 without transient visual impairment, normal visual fields, and with normal anterior segment examination results in both eyes. The fundus of both eyes presented papilledema.

In the neurological examination, the patient was conscious (Glasgow Coma Scale Score of 15) but agitated, without nausea or vomiting; both pupils were normal in size and reactive and she showed no signs of sixth nerve palsy, meningeal irritation, or seizures. Our patient showed moderate (MRC = 4/5) right hemiparesis and right central facial paresis associated with a peculiar type of aphasia.

Language was evaluated using the Romanian version of the Western Aphasia Battery (WAB) [7,8].

The basic language characteristics were as follows. Initially, at the level of conversational language, the patient presented a transitory mutism, followed in the next few hours by non-fluent reduced verbal output and echolalia. Practically, the spontaneous language was absent and speech production was limited to repetition, which was relatively sparse, even for the words without meaning. The repetition was achieved with so much ease that it was best described as echolalia (the immediate, involuntary, and rather accurate repetition of any words spoken by the examiner). When asked a question, she repeated the question verbatim instead of providing an answer. Articulation was good and automatic language was relatively preserved. Pointing and naming were both defective (severe anomia). We observed that auditory and reading language comprehension was severely affected, showing difficulty with passive and complex grammar associated with severely impaired reading (alexia) and agraphia (written expression was severely affected).

Consequently, we noted that she had a severe dual deficit in language comprehension and intentional language formulation, combined with a normal capacity for oral repetition of even long sentences, devoid of any proper comprehension of sentence meaning. We concluded that she presented mixed transcortical aphasia (isolation aphasia).

### 2.2. Imaging

The hyperdense appearance of the straight sinus, Galen vein, and lateral mesencephalic vein on the left revealed of the CT scan performed in ED was considered as a possible sign of cerebral venous thrombosis (Figure 1A,B), as well as venous infarction in the left thalamus, left basal ganglia, left external and internal capsules, and corona radiata, with hemorrhagic transformation in the head and corpus of the left caudate nucleus (Figure 2A,B).

A further CT venography scan was scheduled for a time when the patient was no longer agitated. After admission, the deficit became worse (MRC = 3/5), with stiff neck, anisocoria (right pupil > left pupil), and inability to speak or comprehend others when they spoke, although with preserved repetition (echolalia).

MRI of the brain revealed a left hydrocephalus and the presence of brain edema with compression of the ventricles to the right, as well as a slight hemorrhagic transformation located in the left basal nuclei and thalamus (Figure 3A,B).

Magnetic resonance venography (MRV) of the brain showed almost complete occlusion of the superior sagittal sinus, the straight sinus, the vein of Galen, and in the deep venous system on the left, and only partial occlusion of the lateral sinus and internal jugular vein on the left (Figure 4A–D).

### 2.3. Laboratory Testing: Thrombophilia Testing

Because of the patient’s medical history (onset on the 18th day after a cesarean section for preeclampsia with severe features), clinical spectrum, and the MRI-MRV images, the genetic profiling of thrombophilia was requested and performed, showing severe inherited thrombophilia (with combined thrombophilia represented by the association between homozygous MTHFR C667T, heterozygous Factor V-H1299R, and heterozygous PAI-1 4G/5G).

In addition, serologic measures of thrombophilia were used, including measurement of the anticardiolipin antibodies immunoglobulin G (IgG) and IgM, antigenic protein C, total and free protein S, antithrombin III, lupus anticoagulant, and homocysteine (which were all negative).

The lipid profile showed total cholesterol = 233 mg/dL, LDL = 170 mg/dL, HDL = 37 mg/dL, triglyceride = 102 mg/dL.

### 2.4. Diagnosis

In conclusion, the patient presented focal neurological deficits (consisting of a mixed transcortical aphasia and right hemiparesis) and intracranial hypertension (headache and papilledema), both clinical syndromes due to the postpartum period and to severe inherited thrombophilia affecting multiple cerebral venous and dural sinus thrombosis.

### 2.5. Management in the Acute Phase

The patient was treated with low-molecular-weight heparin (LMWH) in therapeutic dosages (180 anti-factor Xa U/kg/24 h) administered by two daily subcutaneous injections; the appropriate LMWH anticoagulation doses were adjusted based on the blood plasma antifactor Xa level.

She was also treated with osmotic diuretics (mannitol) and hypertonic NaCl 3% sol. for increased intracranial pressure and cerebral edema, along with hydroelectrolytic rebalancing solutions, antihypertensive drugs, painkillers, and prolactin inhibitors.

The patient recovered well and was discharged after 15 days with mild weakness of the upper limb (+4/5MRC) on the right side and anomic plus aphasia (the patient presented the hallmark feature of anomia, i.e., word-finding difficulty, which was noted in spontaneous speech and naming and was associated with circumlocutions and mild impairment of oral comprehension). Circumlocutions consisted of the use of generic terms (e.g., “thing” instead of an intended noun or “boy” instead of a specific boy’s name) and production of fillers such as “like” or “you know”.

### 2.6. Management after the Acute Phase

In order to prevent recurrent CVT and other types of venous thrombosis (deep vein thrombosis or pulmonary embolism), in the patient’s case of severe inherited thrombophilia, we recommended that oral anticoagulation (with apixaban) should be continued after the acute phase of CVT for an indefinite duration.

Clinical and neuroimaging follow-up was performed at 6 months after discharge; the patient presented mild right central facial paresis and no symptoms of aphasia, with partial recanalization of dural sinuses (Figure 5A,B) and hemosiderin deposition in the chronic hemorrhagic area in the head and corpus of the left caudate nucleus and in the left thalamus (Figure 6).

## 3. Discussion

The incidence of CVT is higher in neonates and young adults (especially women) with one or more predisposing factors, including any hypercoagulable state, such as inherited or acquired thrombophilia, pregnancy, puerperium, oral contraceptive, cancer, or recent surgery [9,10].

The predominant genetic mutations linked to CVT are inherited thrombophilias. The three most frequent mutations associated with CVT include factor V Leiden, factor II prothrombin variant (PT 20210A), and the homozygosity for MTHFR C677T [11,12,13]. Women are more predisposed to CVT than men because of hormonal factors, while the highest incidence rate in adults is around 30 years [14,15,16].

The Royal College of Obstetricians and Gynecologists Guidelines list the most important risk factors for venous thrombosis during pregnancy and the puerperium: pre-existing heritable (factor V Leiden) or acquired (antiphospholipid syndrome) thrombophilias; older age (>35 years); obesity (BMI >30 kg/m²); smoking; gross varicose veins; and several obstetric risk factors, such as multifetal pregnancy, pre-eclampsia, cesarean delivery, and preterm birth [14,17].

Pregnancy induces changes in the anticoagulation system that favor clot formation, such as increased levels of fibrinogen and decreased plasminogen and antithrombin III activity. These changes persist throughout the puerperium when the hypercoagulation is favored by decreased blood volume, injuries, immobility, cesarean section, and infections [18,19,20,21]. Moreover, CVT can be caused by pelvic phlebothrombosis via the venous plexuses of the vertebral canal and the basilar venous plexus.

Some of these factors were found in our case, involving a 38-year-old obese female with severe inherited thrombophilia (due to homozygous MTHFR C667T and heterozygous mutations of Factor V-H1299R and PAI-1 4G/5G), with preterm birth at 30 weeks of pregnancy by cesarean section due to early severe preeclampsia 18 days before she was admitted to ED.

The clinical symptoms of CVT are due to two main pathophysiological processes: (1) increased venular and capillary pressure; (2) decreased cerebrospinal (CSF) absorption [5]. Symptoms such as headache, vomiting, papilledema, and altered level of consciousness are caused by the increased intracranial pressure and cerebral edema. The most dangerous consequence of these pathological mechanisms is cerebral herniation because of the increased pressure that affects the nearby tissues [14,22]. We avoided this complication in our case by using osmotic diuretics (mannitol) to treat the augmented intracranial pressure and cerebral edema. In some situations, a brain hemorrhage is caused by increased capillary and venous pressure that leads to the rupture of the vessels and erythrocytes diapedesis through blood–brain barrier damage [23]. This is the explanation for why our patient presented headache and papilledema and had a slight hemorrhagic transformation located in the left basal nuclei and the left thalamus.

CVT can cause a wide spectrum of syndromes starting with isolated intracranial hypertension, focal neurological signs, seizures, and even subacute encephalopathy (from drowsiness to coma); our patient presented symptoms and signs for the first two clinical syndromes. Elevated intracranial pressure can cause visual disorders and papilledema upon ophthalmological investigation (our patient presented only papilledema). The severity of symptoms depends on the particular cerebral veins and dural sinuses, the expansion of brain tissue lesions, and the consequences of the intracranial pressure [14,15,16,17,18].

According to the International Study on Cerebral Vein and Dural Sinus Thrombosis, the most frequently affected dural sinus is the transverse sinus, followed by the superior sagittal sinus [14,15,16]. Transverse sinus thrombosis (which was affected in our case) generally produces headache (as in our patient), and sometimes temporoparietal hemorrhagic infarction (from the obstruction of the vein of Labbé) with hemianopia (which did not occur in our case). If the thrombosis is located on left side of the brain, then it can lead to Wernicke’s aphasia, sometimes with seizures (which was not the case here). Sigmoid sinus (as in our patient) is seldom concerned, but can produce mastoid pain and cranial nerve lesions (V, VI) (which was not the case here). The superior sagittal sinus thrombosis (which did not occur in our case) leads to a variety of symptoms ranging from headache to seizures and focal neurological signs (e.g., hemianesthesia, hemiparesis, visual field defects). Thrombosis of the deep cerebral veins (internal cerebral veins, basal veins of Rosenthal, vein of Galen, straight sinus) (as in our case) is cited in about 18% of CVT patients and produces injuries of the thalami and caudate nucleus. Left lesions produce atypical aphasias and sometimes transcortical aphasias (like in our case), which are difficult to assess due to associated mental status alteration, reduced awareness, or even coma [1,2,5,14,16,22,23,24,25,26].

Our patient had focal neurological signs represented by mild right hemiparesis, central facial paresis, and mixed transcortical aphasia, symptoms that can be explained by the localization of the left cerebral deep venous infarcts that she presented, lesions affecting the left thalamus, caudate, putamen, and periventricular white matter (internal capsule).

Regarding vascular aphasias, the global aphasia (24–38%) and anomic aphasia (20%) are more frequent in acute ischemic stroke. Broca (10–15%) and Wernicke (15–16%) aphasias present intermediate frequencies, while other aphasias are rare [27,28,29,30].

Mixed extrasylvian (transcortical) aphasia is an extremely unusual acute vascular aphasic syndrome (less than 2%), with only a few cases having been reported in the literature [27,28,29,30].

Mixed transcortical (extrasylvian) aphasia is considered an “isolation syndrome” because the spared Broca’s and Wernicke’s areas are completely isolated from the neighboring areas, which are affected [25,26,27]. Consequently, both areas become separated from the other components of the language network, excluding production of spontaneous speech and the comprehension of spoken and written language. For this reason, mixed transcortical aphasia is characterized by a severely diminished quantity of spontaneously generated verbal output, very poor auditory comprehension, and relative sparing of repetition with echolalia. Practically, this type of aphasia combines symptoms of both transcortical motor and sensory aphasia. It resembles a global aphasia with relatively preserved language repetition ability [27,28,29,30].

The differential diagnosis of mixed transcortical aphasia requires this aphasic syndrome to be distinguished from global aphasia and from other forms of transcortical aphasias. Using the Western Aphasia Battery (WAB) test, the diagnosis of any transcortical aphasia requires a relatively normal repetition. In this situation, global aphasia can be ruled out (as in our case), depressed verbal fluency ratings indicate that transcortical sensory aphasia can be excluded (as in our patient), while poor auditory comprehension performance excludes transcortical motor aphasia (as in our case) [27,28,29,30].

The most important cause of mixed transcortical aphasia is represented by cortical lesions isolating the intact perisylvian language areas (watershed zones between the left ACA and MCA in combination with the watershed territory between the left MCA and PCA) (which was not the case here). Usually, consecutive hemodynamic ischemic stroke appears as a result of severe ipsilateral internal carotid artery stenosis [30,31,32,33,34,35].

Another etiology for mixed transcortical aphasias is due to subcortical lesions, including large thalamic hemorrhages interrupting the temporal isthmus and infarcts in the left thalamus, caudate, putamen, and periventricular white matter (as in our case) [32,33,34].

The associated neurological signs (right hemiparesis and central facial paresis) were due to the extension of the lesions in the neighboring periventricular white matter areas (left internal capsule). Hemiparesis (as in our patient) is the most frequent sign in CVT according to multiple case reports [22,23,26].

We did not find any ramifications of vascular occlusion of the ocular veins or arteries (central retinal artery or ciliary posterior arteries) as a thrombotic event associated with the patient’s severe inherited thrombophilia [14].

Various neuroimaging investigation techniques can detect cerebral veins and dural sinus thrombosis. Cerebral edema and venous infarction may be apparent with any modality, although for the detection of the thrombus itself, the most commonly used imaging techniques are computed tomography (CT) and magnetic resonance imaging (MRI), both of which use various types of radiocontrast to perform a venogram and visualize the veins around the brain [13]. Thrombi can be identified using CT or MRI by means of the “dense triangle sign”, the “cord sign”, or the “empty delta sign [3,18,36]. Magnetic resonance venography (MRV) is the preferred and most sensitive diagnostic technique used to identify venous occlusions. Our patient presented an association between multiple dural sinus thromboses (left sigmoid sinus, left and right transverse sinuses, sinus confluence, straight sinus), a left internal jugular vein bulb thrombosis, combined with cerebral vein thromboses (the left deep venous system, including the internal cerebral and basal Rosenthal veins, which unite to form the great vein of Galen).

Many clinical studies have investigated the utilization of anticoagulation to prevent blood clot formation in CVT [37].

Current clinical practice guidelines recommend heparin or low-molecular-weight heparin (LMWH) as the initial and essential treatment, followed by oral anticoagulants (warfarin), unless there are other bleeding risks that make these treatments inappropriate [15,16,17,37,38].

LMWH is associated with substantially lower incidence of heparin-induced thrombocytopenia (HIT) when compared with unfractionated heparin. There is up-to-date evidence confirming that enoxaparin, dalteparin, and tinzaparin are harmless and appropriate for use for the prevention of thrombosis during pregnancy and the puerperal period. After childbirth, it is recommended that each woman be evaluated and that the usage of LMWH be considered if any risk factors are identified. Women in high-risk (as in our case, with severe inherited thrombophilia) groups should prolong thrombosis prevention for 6 weeks with LMWH, while intermediate-risk women should continue treatment for 10 days [15,16,17,37,38].

Oral anticoagulants that are not vitamin K antagonists (NOAC, formerly named as new oral anticoagulants), such as rivaroxaban, apixaban, and dabigatran, can directly suppress factor Xa or the thrombin effect. The FDA do not approve their use during pregnancy; however, if women are not breastfeeding, they can be used after delivery [17]. Our female patient showed significant clinical improvement following LMWH administration and continued apixaban treatment at home.

The outcome of CVT related to the postpartum period is generally a good one, with mortality being under 10% [2,21]. If the patient suspends the usage of anticoagulants, there is a 3% chance of CVT reappearance (which was not the case here for 6 months of follow-up) [5,38].

## 4. Conclusions

The diagnosis of CVST in our case was based on the clinical signs (mixed transcortical aphasia, which is an extremely rare aphasic syndrome), the details of the patient’s obstetrical history, and the significant thrombophilia testing, with magnetic resonance venography–magnetic resonance imaging being used as the gold standard imagery method to identify the thrombi and to understand the extent of the brain damage and cerebral edema.

## Figures and Tables

**Figure 1 diagnostics-11-01425-f001:**
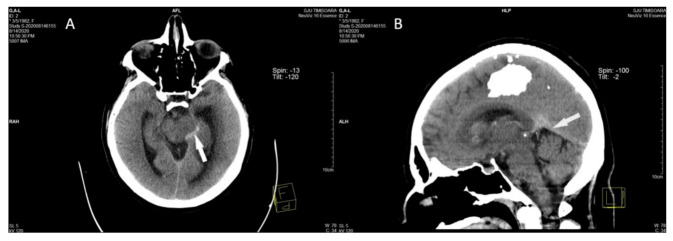
The axial (**A**) and MPR sagittal (**B**) non-contrast head computed tomography performed in the acute phase show the hyperdense appearance (acute thrombosis) of the left lateral mesencephalic vein (arrow in **A**) in the straight sinus and Galen vein (arrow in **B**).

**Figure 2 diagnostics-11-01425-f002:**
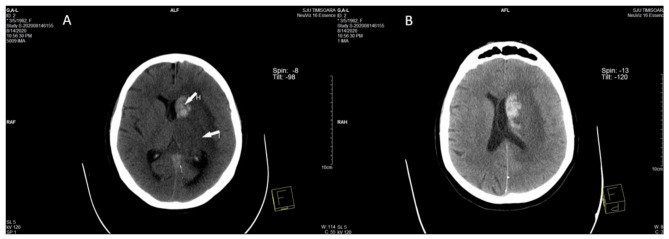
(**A**) Axial unenhanced head CT performed in the acute phase shows venous infarction (I) in the left thalamus, left basal ganglia, left external and internal capsule, and corona radiata. (**A**,**B**) Hemorrhagic transformation (H) can be seen in the head and corpus of the left caudate nucleus.

**Figure 3 diagnostics-11-01425-f003:**
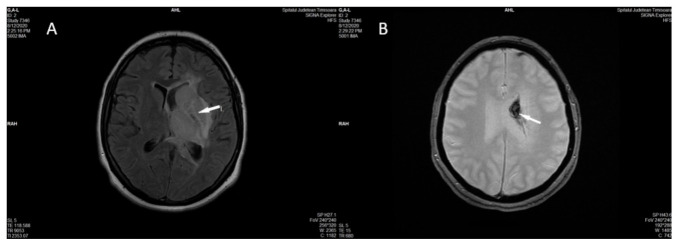
Non-contrast head MRI performed in the acute phase shows venous infarction (I in FLAIR) in the left thalamus, left basal ganglia, and left external and internal capsula (**A**) with hemorrhagic transformation (arrow in Axial T2 Gradient Echo) in the corpus of the left caudate nucleus (**B**).

**Figure 4 diagnostics-11-01425-f004:**
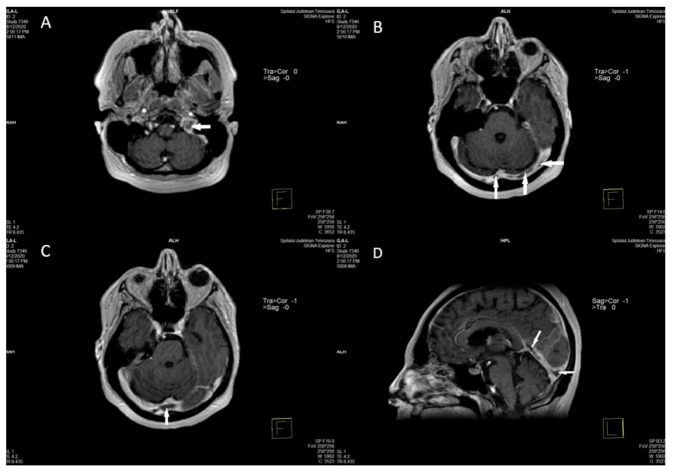
Axial and sagittal T1 postcontrast magnetic resonance images demonstrate extensive filling defects throughout the dural sinuses (arrows—left sigmoid and jugular bulb) (**A**), left and right transvers sinuses (**B**), sinus confluence (**C**), and straight sinus (**D**).

**Figure 5 diagnostics-11-01425-f005:**
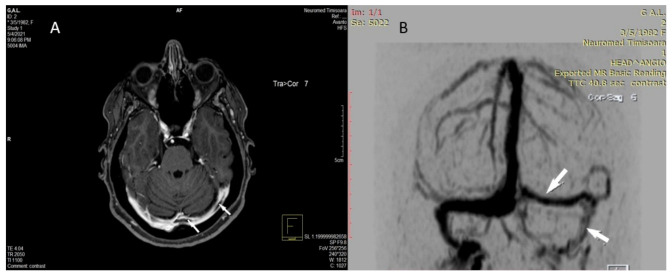
(**A**,**B**) Axial and sagittal T1 MPR postcontrast magnetic resonance images after 6 months, demonstrating persistent partial filling defects throughout the dural sinuses (left transvers sinuses and sinus confluence (**A**), straight sinus (**B**)).

**Figure 6 diagnostics-11-01425-f006:**
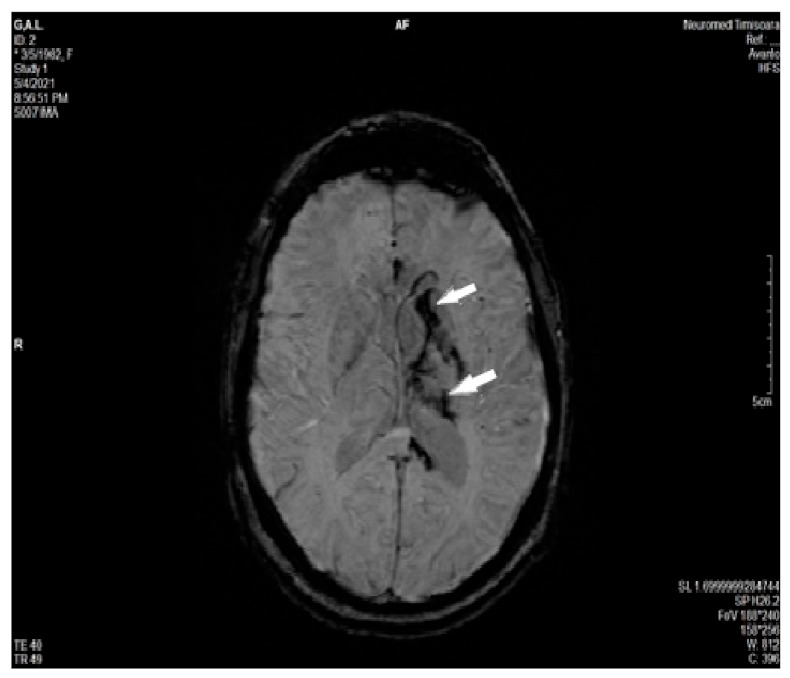
Axial SWI after 6 months demonstrates hemosiderin deposition in the chronic hemorrhagic area in the head and corpus of the left caudate nucleus and in the left thalamus (arrows).

## Data Availability

First Department of Neurology, “Pius Brînzeu” Emergency County Hospital, Timișoara, Romania; Department of Multidetector Computed Tomography and Magnetic Resonance Imaging, Neuromed Diagnostic Imaging Centre, Timișoara, Romania.

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
