# Peer review of "Diagnosis and Management of Mixed Transcortical Aphasia Due to Multiple Predisposing Factors, including Postpartum and Severe Inherited Thrombophilia, Affecting Multiple Cerebral Venous and Dural Sinus Thrombosis: Case Report and Literature Review"

_diagnostics, 2021, doi:10.3390/diagnostics11081425_

Round 1
Reviewer 1 Report
The description of the cerebral venous thrombosis is very good.
I would have described a bit more the trans cortical aphasia and the differences with other types of aphasia. I would also precise the % among all types of aphasia.
Author Response
Thank you for your nice comments and constructive suggestions!
I have attached the answer at your comments in the document below!

Reviewer 2 Report
Very interesting clinical case proposed. Detailed and well-developed arguments in each paragraph. Results of instrumental data exposed well. Images are showed with attention to detail. Award-winning treatment according to the guidelines. Adequate follow up.
Author Response
Dear reviewer, we have made the requested changes. I revised the text and I made the necessary grammatical corrections.
